# Conceptualizing a Gluten-Free Instant Noodle Prototype Using Environmental Sustainability Aspects: A Cross-National Qualitative Study on Thai and Danish Consumers

**DOI:** 10.3390/foods11162437

**Published:** 2022-08-13

**Authors:** Amporn Sae-Eaw, Sasichakorn Wongsaichia, Davide Giacalone, Phaninee Naruetharadhol, Chavis Ketkaew

**Affiliations:** 1Faculty of Technology, Khon Kaen University, Khon Kaen 40002, Thailand; 2International College, Khon Kaen University, Khon Kaen 40002, Thailand; 3Center for Sustainable Innovation and Society, Khon Kaen University, Khon Kaen 40002, Thailand; 4Department of Technology and Innovation, University of Southern Denmark, 5230 Odense, Denmark

**Keywords:** environmental sustainability, instant noodles, gluten-free, sustainable food, sustainable consumption, qualitative study

## Abstract

Gluten-free food products have been developed to satisfy the needs of consumers with celiac disease. However, there has been little research on the product feature development of sustainable gluten-free instant noodles through a qualitative study to explore the customer insights related to environmental attitude and purchase decision. Using a cross-national comparative study between Thai and Danish consumers, this study aims to (1) identify the target customer segments for each country; (2) explore the target customer segments regarding behaviours, desired outcomes, and pain points; and (3) suggest gluten-free instant noodle product prototypes suitable for each country. With a qualitative interview approach, 60 target customers (30 Thai and 30 Danish) were recruited to participate in this research. In addition, a thematic analysis was undertaken to examine their behaviours, desired outcomes, and pain points toward sustainable gluten-free instant noodle products. The findings revealed that convenience-oriented customers were the target segment of gluten-free instant noodle products in Thailand. This segment primarily focused on convenience as the main reason for consuming instant noodles and had common pain points in terms of taste. In contrast, environment-oriented customers were the target customer segment in Denmark. This segment consisted primarily of young women who eat less meat and shared common pain points such as difficulty accessing more sustainable options. Hence, there is a need to educate customers in Thailand (an emerging economy) and increase their awareness regarding environmental sustainability and consumption.

## 1. Introduction

Food systems are crucial to the 2030 Agenda for Sustainable Development, a global commitment to reduce poverty and hunger while minimizing environmental and socioeconomic repercussions [1]. However, food and energy supply chains have complex and entwined environmental and socioeconomic consequences, since changing consumer preferences and consumption patterns pose severe concerns to the overall sustainability of food production and consumption [2]. In particular, convenience food has been widely criticized for harming the environment because it uses a lot of water, energy, and other resources and creates a lot of waste compared to homemade food products, and examples of fast food include soft drinks, processed meat, and instant noodles [3]. However, food is no longer viewed solely as a source of nourishment. Regardless, it has acquired other features because of consumers’ heightened awareness of the different health and environmental consequences of people’s eating habits and food production practices [4]. Therefore, many food companies attempt to generate a lower environmental footprint to mitigate the potential effects of the environment on food production, which promotes ecologically sustainable development in terms of food quality and safety, environmental preservation, and animal welfare [5]. 

Instant noodles, manufactured and distributed worldwide, are a product category that can represent a typical ethnic flavour and are incredibly convenient to consume [6]. According to the World Instant Noodles Association (WINA), the worldwide demand for instant noodles reached 1064.2 million servings in 2019 and has climbed annually since 2015 [7]. Unfortunately, instant noodles have been associated with environmental damage because manufacturers flash-fry instant noodles in palm oil to dry the strands evenly, and palm oil extraction is a significant source of deforestation [8]. Furthermore, most instant noodles are made with wheat flour, which has a negative impact on the environment because wheat requires a lot of fertilizer [9]. Consequently, using more fertilizer can harm the environment because fertilizer releases nitrous oxide gas when it degrades in the soil [9]. For this reason, more research is required to discover more environmentally harmless solutions for instant noodle production processes and products.

There is a growing interest and trend in sustainable consumption regarding gluten-free food products. The global market for gluten-free products was estimated at USD 5.9 billion in 2021 and is projected to expand at a CAGR of 9.8 percent from 2022 to 2030 [10]. The spread of the COVID-19 pandemic has had an explosive effect on the use of gluten-free products due to consumers’ growing health and wellbeing concerns. This growth has been especially noticeable in gluten-free products made of cassavas, which must meet the needs of people with celiac disease. Gluten-free products are perceived to be a healthier and more sustainable solution for modern consumers. The production of gluten-free meals can be more environmentally favourable when done appropriately [11]. Furthermore, individuals with food allergies can benefit from cooking and baking with cassava root because it is gluten-free. Hence, this research proposes a gluten-free instant noodles prototype made of cassavas as an alternative material.

Several research papers studied gluten-free foods and beverages in terms of sustainable consumption [5,11,12,13,14] and focused on customer acceptance, buying behaviour, demand estimation of niche markets, nutritional values, and environmental evaluation [2,5,15]. Based on the previous research, gluten-free instant noodles could still be a novel food product, leaving a research gap in exploring consumer behaviours in a deeper dimension. Furthermore, today, customer experience design for food products is critical to developing new products in a value-creating economy [16]. Understanding consumers’ behaviours, needs, and pain points regarding gluten-free instant noodles consumption would enable producers to create a compelling product experience for consumers. Still, most of the prior work on food product development used sensory evaluation as an instrument to measure, analyze, and interpret behavioural responses. For instance, De Pelsmaeker et al. [17] provided insights concerning the product development of filled chocolates based on customer preferences and sensory attributes. Ruiz-Capillas et al. [18] used novel rapid sensory procedures (check-all-that-apply, napping, flash profile, etc.) to evaluate the quality and marketable feasibility of new meat product development. Rocha et al. [19] utilized qualitative and sensory evaluation methodologies to assess food liking and preference evaluation in children and adolescents based on product characterization and description (for example, expected appearance, taste, smell, and texture). However, these sensory evaluations and conventional qualitative techniques [17,18,19] could not address our research objectives, aiming to identify the fundamental basis of the customers’ pain points. Subsequently, these pain points involve specific problems encountered by current or potential customers, allowing the food product developer to create solution features suitable for the target customer segment [16]. This study adopts a market-centric technique that acknowledges customer insights and reduces ineffective early-stage execution in product development [20,21]. 

Moreover, past studies have not revealed a suitable comprehension of customer needs (desired outcomes) related to product uses, which is a fundamental principle of the lean entrepreneur notion [16]. According to Cooper and Vlaskovits [16], the lean entrepreneur notion contemplates customer behaviours, desired outcomes (needs), and pain points (problems) as the most important factors in identifying potential customers and segmenting customers. Thus, developing a healthy gluten-free instant noodle product prototype could be more feasible when the food companies understand consumers’ behaviours, desired outcomes, needs, and pain points/problems associated with the current instant noodle products. Using the lean entrepreneur notion, a gluten-free instant noodle prototype may fulfill consumers’ needs, solve their issues, and provoke an emotional reaction to attract long-term customers. Morever, this current research employed the segmentation matrix of the lean entrepreneur concept to segment customers based on their needs and pains. As Thailand may represent an emerging market and Denmark may represent a developed world, this study selected Thai and Danish consumers as the sample groups because Asian instant noodle products are popular among young Thai and Danish consumers. As a result of this research, we will be able to suggest a minimum viable product for gluten-free instant noodles representing the desires of the target customers in both Thailand and Denmark, thereby alleviating their pain points and improving customer experiences.

Accordingly, the aim of this study was to (1) explore three representative customer segments from Thailand and Denmark regarding their behaviours, desired outcomes/needs, and pain points/problems regarding instant noodle consumption, (2) identify the target customer segment of gluten-free instant noodles from Thailand and Denmark, and (3) suggest gluten-free instant noodle product prototypes suitable for both Thai and Danish consumers.

## 2. Literature Review

This section discusses the background of instant noodles’ consumption in both Thailand and Denmark. We also introduce related literature on instant noodles consumption in Thailand and Denmark, factors influencing food consumption, gluten-free market trends and the consumption of gluten-free products, and the lean entrepreneur concept.

### 2.1. Consumption of Instant Noodles in Thailand and Denmark

Instant noodles are a ready-to-eat product made from steamed and dehydrated blocks of noodles via deep frying, sachets of powdered broth, and flakes of dry vegetables, meat, or seafood [22]. Instant noodles have gained popularity throughout the world due to their capacity to deliver a meal in a cost-effective, easy, and timely manner [23]. Instant noodles are among the most popular convenience foods among consumers in Asian countries, including China, Korea, Vietnam, and Thailand. In addition, pre-packed or ready-to-eat food products such as instant noodles are popular in some European countries, including Russia, Poland, Ukraine, Hungary, Norway, and the UK [24]. In Denmark, young Danes consume instant noodles as convenient food. Nonetheless, some young Danish consumers eat instant noodles less frequently [19,20] due to their health-conscious lifestyles.

Existing studies related to the consumption of instant noodles have focused mainly on health risk assessment [25,26], nutritional examination, sensory acceptability [27,28], customer attitude [24], and customer behaviour. However, there has been no research on the customer segmentation of gluten-free instant noodles through a qualitative cross-national study. Furthermore, research on the behaviour of instant noodle product consumers has been mostly limited to exploratory research, including the customers’ needs and pain points for instant noodle consumption [9,21]. Studies on instant noodles’ environmental and health concerns influencing consumers’ purchase through a qualitative cross-national study have been few and limited [6]. Moreover, little research studied consumer purchase intentions toward instant noodle products in European countries such as Hungary [24]. Furthermore, a few articles reflect a cross-national study in different countries, including China and Korea, in terms of selection attributes [6]. Therefore, this study addresses this gap with research in two nations with different eating cultures and socioeconomic development: Thailand and Denmark. This study’s originality lies in comparing eating behaviours, needs, and pain points toward gluten-free instant noodles among consumers from two countries (developing vs. developed).

### 2.2. Factors Influencing Food Consumption 

Cha and Wang [6] discovered that flavour and food hygiene was the essential qualities for college students when making instant food selections. Nagy [24] recommended that quality, taste, freshness, and price are critical elements to examine when selecting instant noodles. Bigoin-Gagnan and Lacoste-Badie [29] indicated that food quality, packaging, service, and convenience influenced consumers’ perceived utility value at shoping malls and chain stores, which affected repurchase intention. Taste, color, appearance, smell, brand, price, packaging, and product attractiveness are more essential than health claims when it comes to food purchasing decisions [30,31,32]. Previous research found that consumers value a product’s health benefits when personally relevant to them, such as their health-conscious behaviours and dietary patterns [33]. In contrast, health concerns are associated with organic food consumption and general food consumption [34]. In addition, health is a determining component for young customers, alongside environmental concerns [35]. Lu et al. [36] suggested that millennials are highly responsive to green products and are more educated about the adverse health effects of nonorganic food. As a result, millennials are more committed to green product consumption than previous generations. However, millennials could also be reluctant to purchase organic items due to their higher price [37]. It was found that environmental-oriented customers were willing to alter their consumption behaviors to solve environmetal issues [38].

Additionally, it has been discovered that the nutritional information of food products can influence sensory quality and consumer acceptance [39]. Previous articles claimed that price-sensitive consumers are less concerned with the nutritional composition of products [40]. However, this research indicates that both price and nutrition information are significant determinants of consumers’ food choices—significantly more so than information regarding manufacturers’ social responsibility or environmental impact. In addition, consumers who care about the environment and who are ethically concerned put a lot of value on product qualities that show how food production affects the environment and how food manufacturers treat their workers [40]. Moreover, Vermeir and Verbeke [41] proposed that increasing consumer involvement, perceived effectiveness (of sustainable products), and social peer pressure can all be used to boost sustainable food consumption.

Some studies have examined the effect of national culture on a country’s environmental performance [42,43]. Danish consumers tend to consume organic foods due to health and environmental impact reduction [41]. Swedish customers have high environmental awareness and comprehensive knowledge pertinent to environmental protection [44]. Valentini [45] claimed that individuals from Denmark and Italy previously believed themselves to be environmentally conscious consumers. People continued to prioritize price and overall quality as essential determinants for product selection, but they also began to consider the external consequences of their purchasing decisions. Eco-friendly products and services have been increasingly sought after by consumers, who are willing to pay a premium for goods that are made using environmentally friendly practices [45].

### 2.3. Gluten-Free Market Trends and Consumption of Gluten-Free Products

Gluten-free food products are growing in popularity to meet the demands of people with celiac disease and those who prefer to avoid gluten in their diets. Gluten-free bread products are in high demand due to the increased knowledge and diagnosis of illnesses caused by adverse reactions to wheat, such as wheat allergy, celiac disease, and gluten sensitivity [46]. The market for gluten-free cereal goods is predicted to increase significantly over the next several years, creating numerous product development opportunities for companies looking to sell new gluten-free cereal products that are tasty and inexpensive. Convenience foods, foods with perceived health benefits, low-fat and organic products, range extensions, extending brands, product upgrades, new categories, and premium quality foods are the most crucial food sectors for future NPD development [47]. These areas offer manufacturers the chance to create gluten-free food products that will be accepted by consumers.

Interestingly, prior research has shown that the environmentally friendly nature of gluten-free food products may increase consumption among young adults. A study conducted in Thailand discovered that environmental concern greatly increased young individuals’ willingness to purchase green items [48]. Similarly, a study conducted in Denmark revealed that an attitude of environmental concern influences the purchasing behaviour of well-educated young urban Danish consumers [30]. Consumers have developed a greater awareness of the nutritional value, health benefits, and quality of the foods they consume, and healthiness has become a significant consideration for food purchases [4]. Hence, the demand for healthy convenience foods has increased

Previous research articles studied consumer perceptions of gluten-free products [49,50]. Some studies focus exclusively on reading and comprehending gluten-free product labels; others assess customers’ declarative opinions and consider gluten-free labelling in conjunction with other package features [51,52]. Hence, we observed limited use of different research approaches in the scope of customer perceptions, attitudes, and behaviours. Most research utilized a quantitative approach with a questionnaire survey. However, unlike the quantitative method, the qualitative approach allowed the researcher to explore and better understand complex circumstances regarding customers’ root causes of pain points, which is the fundamental concept of the lean entrepreneur [16]. This study focuses on customer insights regarding behaviors, desired outcomes, and pain points related to gluten-free instant noodle consumption. Recognizing pain points may result in the provision of relevant solutions. Thus, from a business standpoint, it is more viable to use the lean entrepreneur notion to conceptualize sustainable gluten-free instant noodles than to employ a closed-ended questionnaire. 

### 2.4. Lean Entrepreneur: Segmentation Matrix

A segmentation matrix is utilized to reach to the suitable market segment. The segmentation matrix technique enables businesses to find a prospective segment by classifying clients according to need and pain. Purchases are made for a variety of reasons by customers. As a result, it makes more sense to categorize according to how value is created. Customers are classified according to their purchasing habits rather than their demographic characteristics, such as age, gender, and income. Individuals represent distinct market segments depending on their pain level or passion, their expectations of solutions, and their preferred sales techniques. Consequently, the segmentation matrix consists of five indicators to analyze: (1) the level of pain, (2) the budget, (3) the ease of reach, (4) the ease of developing a minimum viable product (MVP), and (5) the market size. Consequently, the segmentation matrix produces a small and well-defined group of people with shared pain or interest.

Traditionally, customer segmentation is a technique that allows researchers and managers to gain a deeper understanding of customers by categorizing them into groups with similar behaviour patterns or attributes [53,54]. By understanding the selection characteristics of each consumer category, food companies will be able to determine which strategies will be most beneficial for each [55]. Previous research typically used a domain-specific instrument for customer segmentation, such as the food-related lifestyle (FRL) for food-related purchasing [56,57,58]. Hence, the findings of these studies recommended that customer segmentation be classified into three segments: convenience-oriented consumers, health-oriented consumers, and environment-oriented customers [59,60,61].

### 2.5. Personas and Zoom Tools for Lean Entrepreneur

The lean entrepreneur approach ensures that products are designed to engage customers. A customer persona is a semi-fictional figure developed from user research data that represents the important features of a subset of buyers. Consequently, this instrument investigates the demands of actual users by getting insights into their traits, habits, needs, and pain points. It helps to gain a better knowledge of clients, benefiting product development and marketing activities. In contrast to conventional marketing and sales, the lean entrepreneur approach stresses customer relationships, attractiveness, and problem-solving. Additionally, it is preferable to select a customer segment that fits the profile of the firm’s ideal customers [20,21]. 

Moreover, the startup should utilize the customer, problem, and solution zoom tools to enhance the personas. Zoom tools are used to ascertain what truly motivates consumers. It is necessary to zoom in on unique persons rather than general personas to understand customer emotions properly. First, the customer zoom tool enables businesses to define the characteristics and behaviours of real individuals better to understand their requirements, desires, and aspirations. After focusing on clients, the organization should concentrate its focus on specific challenges or demands. Second, the problem zoom tool permits the business to clarify the client’s hurdles and specific goals. Then, a minimal viable product (MVP) should be created in order to identify the precise features that will add benefits for users or for customers. At this point, the organization should speculate on the requirements for the MVP. Finally, the solution zoom tool enables businesses to identify which components of their design should be tested before actual development.

### 2.6. Study Context and Research Framework

From the researchers’ perspective, regular instant noodles consumers are the target customers of the gluten-free instant noodles prototype. In this context, participants are both non-celiac gluten sensitive and gluten intolerant people in Thailand and Denmark. Hence, the lean entrepreneur concept necessitates a qualitative study to identify consumers’ insights, including their behaviors, viewpoints, desired outcomes, pain points, and previous experiences with instant noodles [16]. Additionally, consumer insights helps to develop a thorough understanding of how consumers feel and think about sustainable gluten-free instant noodles. This research has the ability to increase customer empathy by uncovering customer insights rather than simply understanding what the consumer wants. Subsequently, this study combined a qualitative method, specifically thematic analysis, with the lean entrepreneur concept before creating a gluten-free instant noodle prototype for the suitable customer segment. Figure 1 depicts the overall framework for this research. The first step, known as the customer empathy stage, is to understand the customers’ behaviour, desired outcomes, and pain points. The segment selection stage is the second step in finding the most likely consumer for sustainable gluten-free instant noodles. The solution design stage is the third and last step in the procedure.

## 3. Research Methodology

### 3.1. Participant Recruitment 

This study interviewed consumers who had eaten instant noodles in two countries (Thailand and Denmark). Participants (*n* = 60; Thai = 30 and Danish = 30) were recruited from the six most populated provinces in the Northeastern Thailand (Nakornratchasima, Ubonratchathani, Khon Kaen, Udonthani, Buriram, and Srisaket) and Odense, Denmark. We decided to interview ten individuals for each segment in the two nations. It is generally accepted that a sample size of 25 to 30 is sufficient for most qualitative studies [62,63,64]. This resesarch met the criteria of the Exemption Determination Regulations by Khon Kaen University’s Ethics Committee for Human Reseasrch (KKUEC). Based on the KKUEC’s regulation, we showed the document explaining the interview procedures to the participants including their rights, and informed consent was obtained from all participants. We conducted face-to-face interviews at the public areas in the stated provinces, for instance, parks, local markets, and parking areas of convenience stores and supermarkets. The recruited participants are observed based on the inclusion and exclusion criteria, which help to determine the actual instant noodle consumers. Hence, gluten-free instant noodles products’ inclusion and exclusion criteria were categorized based on the instant noodles consumers’ character traits and behaviours. In addition, based on the research ethics standard, the inclusion criteria were consumers 20 years of age and older. In contrast, we excluded individuals under 20 years of age or those who had never eaten instant noodles.

### 3.2. Script Development and Data Collection

Semi-structured interviews were employed to gain comprehensive data on participants’ opinions and experiences with gluten-free instant noodles [16]. Audio-recorded interviews, lasting no more than 10–15 min, were conducted in Thai and English and led by trained interviewers. The details of the study were communicated to the participants. They were given a document assuring them of the research’s ethical principles, including anonymity and confidentiality. They might terminate the interview or refuse to answer questions whenever they deemed it necessary before beginning the session. In addition, the purposive sampling approach was used in data collection via a semi-structured interview. The purposive sampling method was generally utilized based on the fundamental purpose and objective of the study; additionally, the sampling design is flexible and emerges during the analyses in qualitative research [65]. The purposive sampling technique was employed since only people that consumed instant noodles in Northeast Thailand and Odense, Denmark, understood the issues under investigation.

As presented in Figure 2, prior to the interviews, consumers were divided into three main proposed segments, including convenience-oriented customers, health-oriented customers, and environment-oriented customers. The proposed customer segments corresponded to the instant noodle consumers’ personas, which were segmented primarily by their behaviour, desired outcomes, and pain points. 

Moreover, pilot tests were also performed to ascertain that the three instant noodle customer segments represented the prosepctive customers in the market. The surveyors interviewed three participants of each customer segment in the six most populated provinces in the Northeastern Region of Thailand and Odense, Denmark. The summarized customer segments were grouped into three groups based on common characteristics, needs, and pain areas among consumers in each group. As previously stated, the three customer segments considered were convenience-oriented customers, health-oriented customers, and environment-oriented customers.

Consumers were assigned to each of the three customer segments using the segmentation matrix illustrated in Table 1. This matrix rates the ratings based on factors such as the depth of pain, the budget, the ease of reach, the ease of MVP, and the size of the market [16]. To assess those requirements, we developed interview questions based on zoom tools that elicited information about consumers’ behaviours, desired outcomes, and pain points. These interview questions consist of semi-structured interview questions that incorporate checklists, open-ended questions, and rating scales. The interview questions consist of two sections: (1) demographic characteristics of consumers, which are related to the unique attributes of the customer zoom tool (gender, age, education level, occupation, income, and purchase frequency); and (2) purchasing and eating behaviours of respondents, which related to the behaviours from the customer zoom tool (See Section A.1 and Section A.2). 

The participants then rated the intensity of their pain on a scale ranging from mild to severe (L = low, M = medium, and H = high), where H = 3 points, M = 2 points, and L = 1 point. The budget ratings for low, medium, and high were calculated using the participants’ monthly income. In addition, we developed questions regarding the ease of customer interaction and the speed and consistency with which items can reach customers to determine reachability requirements. By adopting low, medium, and high levels of accessibility, researchers could determine which level suited their research objectives the best. The solution zoom instrument was also utilized to evaluate the MVP’s usability by analyzing the list of features and the minimum effort required to develop solution functionality [16]. We then determined the minimum viable product and ranked the alternatives on low, medium, and high scales. Lastly, secondary data from prior statistical information on the target customers were utilized to determine the size of the market. Due to this, low, medium, and high scores were evaluated independently. Finally, we calculated the average score for each criterion in Table 1 by segment, added the total score for each segment, and ranked each segment as follows: 1, 2, 3 = segment with the highest score, second highest score, and lowest score, respectively.

### 3.3. Data Analysis

All interviews were audiotaped and transcribed verbatim by a team of professional transcriptionists. The results of the thematic analysis were utilized to determine common themes in the replies and their relationship to the study topics [66]. The interviews were initially coded thematically with the assistance of customer and problem zoom tools, utilizing a deductive coding method [16]. The coding strategy was changed and enhanced inductively when more subjects were identified throughout the investigation [67]. NVivo version 12, a program for qualitative data analysis, was used to help organize the data.

## 4. Results

Table 2 summarizes the demographic characteristics of the participants. Four male and six female interviewees participated in the convenience-oriented customer segment for Thai participants. The majority of participants were between the ages of 20 and 30. They had bachelor’s degrees. Most of them were students and earned less than $1000 per month. The majority of respondents bought instant noodles three to four times per week.

In addition, in the Thai health-oriented customer segment, most participants were female. The majority of participants were between 51 and 60, which accounted for six interviewees. Many respondents had completed bachelor’s degrees. Most of them were public sector employees and earned less than $1000 per month. The majority of responses reported purchasing instant noodles less than once per week.

In contrast, most of the respondents were female, which accounted for seven of the interviewees in the environment-oriented customer segment. The majority of participants were between the ages of 31 and 40. Many of them possessed bachelor’s degrees. They were private-sector employees and were paid 2001–3000 USD per month. The majority of respondents reported purchasing instant noodles less than once per week.

For Danish participants, five male and five female interviewees participated in the convenience-oriented customer segment. The majority of participants were between the ages of 20 and 30. They had completed their bachelor’s and master’s degrees. Most of them were students and earned less than $1000 per month. The majority of participants purchased instant noodles less than once per week.

Moreover, in the health-oriented customer segment, the majority of respondents were female. Most of the participants were between the ages of 31 and 40. They tended to have doctoral degrees, were private-sector employees, and earned $4001–5000 USD per month. Every participant bought instant noodles less than once per week.

On the other hand, there were female interviewees in the environment-oriented customer segment. Five interviewees were between the ages of 20 and 30, which accounted for the majority of respondents. They had completed doctoral degrees, worked as university students, and earned more than 5000 USD per month. All of them stated that they purchased instant noodles less than once a week. The data related to the procedures of interviews for both customer segments including locations, interview dates, and durations are stated in Appendix B.

### 4.1. Thai Consumer: Convenience-Oriented Customer

#### 4.1.1. Behaviors

##### Theme: Convenient Eating Habits

In Thailand, convenience-oriented customers often purchase instant noodles as their quick meals due to the little time involved in their preparation. In addition, this segment usually buys other convenience foods and uses online food delivery applications because they would like to minimize the time and effort involved in food preparation. Furthermore, some of them do not normally cook. Hence, they frequently purchase food from supermarkets, restaurants, and local food shops. Ten of the respondents replied as follows:

“I usually eat instant noodles 2–3 times per week. I cook instant noodles in the microwave because it is convenient for me.” (Interviewee no.1)

“I am not good at cooking. So, I prefer to buy food from local food shops or restaurants nearby my house.” (Interviewee no. 2)

“I love eating Kuey Jab Yuen (Vietnamese noodle soup) so much. However, I do not know how to cook this menu. Hence, I normally buy Kuey Jab Yuen from the noodle shops right on the corner.” (Interviewee no. 3)

“I often eat noodle soups twice a week. I normally order noodle soups by using food delivery apps since it can save me a lot of time.” (Interviewee no. 4)

“I am fond of eating any kind of noodles. Most of them are instant noodles because they are very delicious and easy to eat. I also like Kuey Jab Yuen (Vietnamese noodle soup) because my grandmother used to cook these noodles when I was younger.” (Interviewee no. 5)

“Noodles have been one of my favorite dishes since I was a child. As long as I can remember, my mom has always cooked homemade noodle soups for me and my sister. However, if I want to eat some noodle soup, I might order from well-known noodle soup shops nearby where I live.” (Interviewee no. 6)

“I sometimes buy ready-to-eat Vietnamese noodle soups from the online shopping apps because I think it must be easier to start cooking noodles from scratch.” (Interviewee no. 7)

“I mostly buy breakfast from convenience stores. But, if I have time to eat breakfast at the local food shops, I will probably order Kuey Jab Yuen (Vietnamese noodle soup) because it makes me feel nostalgic.” (Interviewee no. 8)

“I do not know how to cook noodle soups. Thus, I usually buy from food canteens or order from any shops that are recommended in online food delivery apps.” (Interviewee no. 9)

“I would probably buy ready-to-eat foods from convenience stores because it could save me time cooking in the morning.” (Interviewee no. 10)

#### 4.1.2. Desired Outcomes

##### Theme: Sticky and Elastic Noodles

The findings revealed that the convenience-oriented customer segment desires the sticky and elastic textures of noodles. All of them prefer soft and sticky noodles that save time cooking and melt in their mouths. Ten of the respondents replied as follows:

“I want soft and smooth noodles. Some brands take some time to cook until the noodles become soft.” (Interviewee no. 1)

“It would be good to have soft noodles. I want the noodles to melt in your mouth.” (Interviewee no. 2)

“I like tiny noodles because I eat and chew them for a short time. I also like the sweet taste in the soup.” (Interviewee no. 3)

“I am fond of soft and sticky noodles with natural color.” (Interviewee no. 4)

“I like the proper size, the same as the spaghetti noodles. It would be great to have a soft and smooth texture with an umami taste.” (Interviewee no. 5)

“The noodles should be elastic and soft, not stick together too much because it is not easy to cook and eat.” (Interviewee no. 6)

“I love thick noodles, not the thin ones.” (Interviewee no. 7)

“I would like to have soft and sticky noodles that are hard to tear off.” (Interviewee no. 8)

“In my opinion, I prefer soft and sticky noodles with attractive colors.” (Interviewee no. 9)

“I want to say that instant noodles are probably not too big or too small. I mean, it should be the proper size when I eat instant noodles.” (Interviewee no. 10)

#### 4.1.3. Pain Points

##### Theme: Taste Dissatisfaction 

The results indicated that the convenience-oriented customer segment tends to have pain points regarding the taste satisfaction of instant noodles. They were more likely to select the flavors according to their preferences. Most of them stated that there were fewer alternatives to flavored instant noodles. Seven of the respondents replied as follows:

“I expect that gluten-free instant noodles will have several flavors to select from in the supermarkets.” (Interviewee no. 1)

“The flavor will probably be the pain point due to fewer options.” (Interviewee no. 2)

“I wish there would be various flavors of instant noodles (guay jab yuan).” (Interviewee no. 3)

“I would rather buy the spicy taste of instant noodles, but some of them are way too spicy for me.” (Interviewee no. 4)

“I think some brands of instant noodles’ seasoning are not umami enough for my preferences.” (Interviewee no. 5)

“The flavor of the noodles is probably the pain point, because I want to try charcoal instant noodles.” (Interviewee no. 7)

“The taste of gluten-free foods can be a pain point because I used to try gluten-free pasta, but it is not as delicious as typical pasta.” (Interviewee no. 10)

### 4.2. Thai Consumer: Health-Oriented Customer

#### 4.2.1. Behaviors

##### Theme: Healthy Eating Habits

In Thailand, the health-oriented customer segment rarely consumes convenience foods, including instant noodles, due to their unhealthy ingredients. They were more likely to consume fresh organic produce due to its chemical-free substance content. If they must eat instant noodles, they would rather add other raw ingredients such as vegetables and meat to increase nutrients. Ten of the respondents replied as follows:

“In the past, I ate a lot of unhealthy foods. However, I stopped eating those kinds of foods and am eating a healthier diet.” (Interviewee no. 11)

“I try to balance my diets. For example, if today I eat too many processed foods, the next day I will eat more fresh raw foods and vegetables.” (Interviewee no. 12)

“I have lactose intolerance. When I drank milk, I felt uncomfortable in my stomach. So, I stopped drinking them for a while. Moreover, I tend to be aware of what I am going to eat because of the current allergic condition that I have now.” (Interviewee no. 13)

“I try to eat every food in moderate amounts because it is not good for my body to eat too much or less of certain foods. So, I eat instant noodles sometimes, probably once a month. I love to add vegetables that I like on top of instant noodles.” (Interviewee no. 14)

“I rarely eat instant noodles. The last time I ate them was when I traveled to Japan. The reason why I rarely eat them is that I think they are not healthy, and I can eat other foods that are much healthier.” (Interviewee no. 15)

“I’m a vegetarian because of my parents, but I think it’s good for my digestive system because I probably can’t eat as much meat as I used to.” (Interviewee no. 16)

“I am allergic to dairy. Hence, I am unable to eat any dairy products anymore. This causes me a lot of anxiety when I decide to eat another food because I am afraid of getting allergic again. Besides, I tend to eat less meat at meals.” (Interviewee no. 17)

“I used to eat a lot of processed foods, including instant noodles, but I quit eating those foods since I was diagnosed with breast cancer.” (Interviewee no. 18)

“I tend to select organic foods from the local farms because I can ensure that the vegetables and fruits will be safe.” (Interviewee no. 19)

“In my opinion, I think instant noodles are bad for my body because they contain preservatives to make them last longer.” (Interviewee no. 20)

#### 4.2.2. Desired Outcomes

##### Theme: Nutritional Value in Noodles

The findings showed that this segment wanted instant noodles with nutrient-rich sources for their quick healthy meals. Most of them required vegetables as an ingredient in making noodles themselves. Some of the participants preferred low calorie, low sodium, and no MSG foods. Ten of the respondents replied as follows:

“I want more healthy instant noodles, such as low sodium.” (Interviewee no. 11)

“I prefer to have a veggie flavor because I think it could be healthier than meat flavors.” (Interviewee no.12)

“I would love to eat instant noodles with low calories because I am on a diet.” (Interviewee no. 13)

“I love eating vegetables due to their high fiber and nutrients. So, I wish to have very healthy instant noodles with good taste.” (Interviewee no. 14)

“I want to control my weight. So, I would like to eat low-calorie and low-sodium foods. If instant noodles could have more nutrients, which would be great for me.” (Interviewee no. 15)

“If they had vegetable instant noodles, I would eat them more often because I love vegetables.” (Interviewee no. 16)

“I like instant noodles because they have low sodium and no MSG. Additionally, the instant noodles may have high fiber content inside.” (Interviewee no. 17)

“It might be good for me to have healthy instant noodles with less sodium, less salt, and less MSG.” (Interviewee no. 18)

“I think the noodles should have other nutrients instead of only the carbohydrates from the flour.” (Interviewee no. 19)

“I prefer low-calorie instant noodles with high fiber content. So, I can enjoy eating my favorite noodles and have some good health benefits at the same time.” (Interviewee no. 20)

#### 4.2.3. Pain Points

##### Theme: Food Quality and Safety

The results demonstrated that the most common pain points in this segment were lack of quality and safety regarding instant noodles’ ingredients. In addition, instant noodles consist of preservatives that help instant noodles last longer. Hence, they assumed that instant noodles could be harmful to their health status. Seven of the respondents replied as follows:

“The obstacle is the unhealthy ingredients in instant noodles.” (Interviewee no. 12)

“I think preservatives are the important issue. That is why I rarely eat instant noodles.” (Interviewee no. 13)

“I found out that I gained weight after eating instant noodles. I think it is probably because of the high sodium.” (Interviewee no. 14)

“There are unsafe food additives in instant noodles. This is the main pain point that I worry about when purchasing any kind of food.” (Interviewee no. 15)

“I do not like highly processed foods due to the unhealthy food content.” (Interviewee no. 16)

“I do not trust the food producers regarding food safety.” (Interviewee no. 18)

“I have a hard time selecting foods due to ingredient quality.” (Interviewee no. 19)

### 4.3. Thai Consumer: Environment-Oriented Customer

#### 4.3.1. Behaviors

##### Theme: Peer Pressure and Increased Environmental Concerns

In Thailand, the environment-oriented customer segment rarely purchases processed foods that have an environmental impact on carbon footprint emissions. Moreover, most of the participants became environmentally conscious because of peer pressure. For instance, some of them stated that their friends were the main factors that influenced them to be more concerned about food and environmental harmfulness. Ten of the respondents replied as follows:

“Since I realize that food production pollutes the environment, I have quit eating processed foods and use eco-friendly food containers. I started to become a pro-environmentalist because my best friend encouraged me to do so.” (Interviewee no. 21)

“My friends have a big influence on me changing my diet based on natural and environmentally friendly principles.” (Interviewee no. 22)

“We have a small group of people who are concerned about the environment. So, this can encourage me a lot when choosing foods.” (Interviewee no. 23)

“I used to study abroad, and I changed my behaviors and attitudes because of my classmates and flatmates. I started to eat more organic foods to reduce my carbon footprint.” (Interviewee no. 24)

“I have been trying to use fewer foam and plastic food containers since I joined the pro-environmental club.” (Interviewee no. 25)

“I would say my work collages inspire me to be more concerned about the environmental impacts of my food selections.” (Interviewee no. 26)

“My peers are the ones who persuade me to eat organic foods and use fewer plastic bags.” (Interviewee no. 27)

“I personally think that environmental awareness is guided by the community. Due to increased carbon footprints, I eat fewer processed foods now.” (Interviewee no. 28)

“I became a vegetarian since I realized that meat production can harm the environment.” (Interviewee no. 29)

“My friends play an important role in motivating me toward sustainable consumption. After that, I stopped eating frozen foods or other ready-to-eat foods due to health and environmental effects.” (Interviewee no. 30)

#### 4.3.2. Desired Outcomes

##### Theme: More Channel Shopping for Sustainable Foods

The results implied that this segment needed multiple shopping channels for sustainable foods that were environmentally friendly, including gluten-free instant noodles. In addition, they required ready-to-eat foods that are less harmful to the environment to be sold in both online and physical stores. Ten of the respondents replied as follows:

“It would be nice if there were more organic food shops in this town. I would like to have ready-to-eat foods like instant noodles as well. However, I hope gluten-free instant noodles will be more environmentally friendly than ordinary instant noodles.” (Interviewee no. 21)

“There are only a few sustainable grocery shops here. So, I would expect to have more places where I can buy eco-friendly food.” (Interviewee no. 22)

“It is hard to find gluten-free food products where I live. It would be more convenient if there were more gluten-free food products in local food stores.” (Interviewee no. 23)

“Since I only see one brand in the market, I would rather buy various choices of sustainable instant noodles.” (Interviewee no. 24)

“I anticipated that there would be more brands of instant noodles that use eco-friendly packaging because I would be willing to support such a brand.” (Interviewee no. 25)

“I actively desire more instant noodles that are healthy and sustainably made. As we all know, processed food production can damage the environment.” (Interviewee no. 26)

“I wish to have processed foods contained in biodegradable packaging.” (Interviewee no. 27)

“I would like food companies to start using more eco-friendly food containers, which are good for our planet.” (Interviewee no. 28)

“I think my desired outcome is to have more gluten-free foods in the local supermarkets near my home.” (Interviewee no. 29)

“I need frozen foods in eco-friendly packaging.” (Interviewee no. 30)

#### 4.3.3. Pain Points

##### Theme: The High Price of Eco-Friendly Foods

The findings revealed that the participants of this segment had product cost pain points towards gluten-free food products and eco-friendly diets. All of them claimed that the majority of gluten-free and sustainable food prices were higher than those of gluten-filled and eco-unfriendly foods. Ten of the respondents replied as follows:

“Well, I think gluten-free diets are more expensive than typical diets.” (Interviewee no. 21)

“I found that the price of organic produce is much higher than non-organic produce.” (Interviewee no. 22)

“I spend lots of money buying gluten-free foods from online stores. So, this could be one of the pain points.” (Interviewee no. 23)

“The price point could be the higher price of sustainable foods.” (Interviewee no. 24)

“Nevertheless, I hope the price is reasonable enough to purchase.” (Interviewee no. 25)

“Price is one factor that I consider before shopping for food.” (Interviewee no. 26)

“But I have seen many bridgeable packaging brands that are pretty expensive when compared with foam food containers.” (Interviewee no. 27)

“I want to say that biodegradable packaging is more costly than plastic food containers.” (Interviewee no. 28)

“However, the price is a little bit higher than gluten-free foods.” (Interviewee no. 29)

“My obstacle is the cost of eco-friendly packaging.” (Interviewee no. 30)

### 4.4. Danish Consumer: Convenience-Oriented Customer

#### 4.4.1. Behaviors

##### Theme: Convenient Eating Habits

In Denmark, the convenience-oriented customer segment usually eats bread, sandwiches, cinnamon rolls, and other bakery products. However, some of them have eaten instant noodles as their quick meals from time to time. Ten of the respondents replied as follows:

“Well, instant foods consume less time to cook, which is good for a short time.” (Interviewee no. 31)

“I eat instant noodles when I feel too lazy to cook, but I mostly eat ready-to-eat pasta that is sold in supermarkets.” (Interviewee no. 32)

“Last month, during our vacation trip, I bought a cup of instant noodles. I simply want to shorten the time it takes to prepare meals.” (Interviewee no. 33)

“I normally eat instant noodles when I am in a rush hour.” (Interviewee no. 34)

“Honestly, I’ve eaten instant noodles a few times when I was travelling in the city.” (Interviewee no. 35)

“I love eating chicken instant noodle soup because it is fast for me to cook.” (Interviewee no. 36)

“I used to eat instant noodles when I was in university because I did not have time to cook.” (Interviewee no. 37)

“If I want to eat something quickly, I will eat instant noodle cups. I am also not good at cooking. So, I would rather buy convenience food for every meal, such as frozen creamy mushroom pasta or BBQ hot wings.” (Interviewee no. 38)

“I prefer to eat ready-to-eat foods from time to time, especially when I need food during a busy day.” (Interviewee no. 39)

“My main dishes are bread, pasta, and bananas. It is fast prepared in the morning. However, I tried eating reman instant noodles a few times. They were delicious.”  (Interviewee no. 40)

#### 4.4.2. Desired Outcomes

##### Theme: Softness Texture of Noodles

The results indicated that the convenience-oriented customer segment in Denmark required the soft texture of noodles to satisfy their preferences. Five of the respondents replied as follows:

“I would like the noodles to be soft since I like the way the noodles melt in my mouth.”  (Interviewee no. 32)

“I expect gluten-free instant noodles to be as soft as ordinary instant noodles.”  (Interviewee no. 33)

“I used to eat very softly reman noodles in Japan. Hence, I wish to eat very soft instant noodles in supermarkets here as well.”  (Interviewee no. 37)

“I would rather have soft and thick noodles with sweet soup.”  (Interviewee no. 38)

“I prefer to cook instant noodles that get mushy very quickly.” (Interviewee no. 40) 

#### 4.4.3. Pain Points

##### Theme: Spiciness

The findings showed that this segment had common pain points regarding the level of spiciness in instant noodles. Most of them expressed that the consumption of spicy foods can cause stomach conditions. Eight of the respondents replied as follows:

“I cannot eat the spicy taste of instant noodles. So, this issue must be the pain point.” (Interviewee no. 32)

“Well, I find that my tongue and mouth cannot tolerate spicy foods. So, it can be an issue for me when I eat spicy instant noodles.” (Interviewee no. 33) 

“I can eat spicy foods, but not too spicy.”  (Interviewee no. 34)

“I think many Asian instant noodles are way too spicy for me.”  (Interviewee no. 35)

“I enjoy eating spicy foods, but I think some instant noodles are too spicy for me.” (Interviewee no. 36) 

“After I eat spicy Korean instant noodles, I feel uncomfortable in my stomach.” (Interviewee no. 37) 

“I struggle with eating chili garlic ramen noodles because they are too spicy.”  (Interviewee no. 38)

“I think the level of spiciness is the problem. I was sweating every time I ate Korean instant noodles.”  (Interviewee no. 40)

### 4.5. Danish Consumer: Health-Oriented Customer

#### 4.5.1. Behaviors

##### Theme: Healthy Eating Habits

In Denmark, the health-oriented customer segment rarely consumes processed foods such as breakfast cereals, instant noodles, sausages, tinned vegetables, etc., because of their unhealthy levels of added sugar, sodium, and fat. People there are more concerned about their diets in terms of nutrients and health benefits. Ten of the respondents replied as follows:

“I want to balance my diet by eating more vegetables and eating less meat. Sometimes, I eat plant-based meat instead of raw meat. I sort of like instant noodles as well. So, when I cook instant noodles, I will add vegetables and plant-based meat.” (Interviewee no. 41) 

“I have changed my eating habits quite a lot since I started to realize that I am getting older, and my digestive system is not as good as it used to be in the old days.” So, I started researching healthy eating diets and tried to eat less meat and processed foods. However, I still eat instant noodles from time to time.” (Interviewee no. 42) 

“I do try to cut down on processed foods as much as I can. I used to store a lot of instant noodles and sausages in the fridge, but right now I do not have any instant noodles left at home.”  (Interviewee no. 43)

“I have become vegan since I got ill two years ago. So, I am stopping eating any kind of meat and eating more vegetables and fruits.”  (Interviewee no. 44)

“Most of the food ingredients that I consume are organically produced. I usually buy fresh organic vegetables and fruits from the local farms that I know. I have to check the organic certificates of the farms before I decide to buy their foods.”  (Interviewee no. 45)

“I think I consider myself a health-conscious person because I am aware of what I am eating. I rarely eat instant noodles due to the high sodium content. However, I would probably be interested in healthy instant noodles if they are available on the market.” (Interviewee no. 46)

“I tend to eat more organic foods since I am attempting to maintain my health. I would probably go for the less processed food instead of instant noodles.”  (Interviewee no. 47)

“I used to eat a lot of instant noodles in the past, but I stopped eating them for a long time because of the unhealthy ingredients.”  (Interviewee no. 48)

“I try and balance out a week with different diets; hence, it would be more veggies, fruits, and some meat alternatives. Due to my health condition, I cut down on eating red meat for a while.”  (Interviewee no. 49)

“I have been a vegetarian for 10 years since I realized that my stomach does not work well with meat. So, if I want to get some instant noodles, I will probably choose the veggie instant noodles.”  (Interviewee no. 50)

#### 4.5.2. Desired Outcomes

##### Theme: Nutritional Values in Noodles

The findings of the study revealed that this segment wanted instant noodles with the proper nutrients for their well-being. The participants considered nutritional values to be the important factors when making food choice decisions. Ten of the respondents replied as follows:

“I enjoy eating organic and fresh foods like vegetables. So, I desire to have some ready-to-eat veggie foods that are healthy and convenient for me.”  (Interviewee no. 41)

“I prefer to eat foods with high fiber content, which is good for my digestive system.” (Interviewee no. 42) 

“Nowadays, I try not to eat processed foods because they usually contain ingredients that could be harmful to my health. Thus, I would like to have healthier gluten-free instant noodles that contain nutrients.”  (Interviewee no. 43)

“When I make a purchasing decision, I will consider the nutrients contained in those foods. Hence, if instant noodles increase some nutritional values, I probably would like to try them.” (Interviewee no. 44) 

“I am concerned about the nutrients from the foods that I am going to eat. It would be nice if instant noodles contained high fibers that were good for the digestive system as well.”  (Interviewee no. 45)

“I am very careful about what I eat. So, I wish to have gluten-free instant noodles that have nutrients.” (Interviewee no. 46) 

“I would like to try instant noodles made from vegetables such as spinach, cabbage, and carrots.”  (Interviewee no. 47)

“I do not eat instant noodles because of MSG. If there are non-MSG-containing instant noodles, I would like to buy them.”  (Interviewee no. 48)

“I am not allergic to sodium or MSG, but I try to avoid eating them due to their health impacts. I think I would like to eat more healthy instant noodles, such as low carb, low sugar, and low sodium. Moreover, instant noodles should have some nutrients from natural ingredients.”  (Interviewee no. 49)

“I want more veggie instant noodles in the supermarkets because I am vegetarian, and I want to have healthier ingredients in convenience foods like instant noodles.”  (Interviewee no. 50)

#### 4.5.3. Pain Points

##### Theme: High Sodium Content

The results of the study showed that this segment had pain points in terms of the high sodium content in instant noodles. They mentioned that diets with high sodium could increase the risk of developing high blood pressure, which is a significant cause of stroke and heart disease. Ten of the respondents replied as follows:

“One of the reasons I do not eat instant noodles is their high sodium content.”  (Interviewee no. 41)

“I felt like the instant noodles were too salty. So, I try not to eat very often.” (Interviewee no. 42) 

“The problem is that instant noodles have high sodium. I stopped eating foods with high sodium because sodium can harm my health, such as blood pressure or heart attack.” (Interviewee no. 43) 

“The pain point is the sodium in instant noodles.”  (Interviewee no. 44)

“I would like to have low sodium instant noodles, which is the main issue that I have now.” (Interviewee no. 45) 

“I think there is too much sodium and preservatives in instant noodles. Hence, this could be a pain point from consuming ordinary instant noodles.”  (Interviewee no. 46)

“I want to say that high sodium and MSG are the pain points.”  (Interviewee no. 47)

“I would rather have low sodium content in instant noodles.” (Interviewee no. 48) 

“From my perspective, the only problem with eating instant noodles is sodium.”  (Interviewee no. 49)

“Well, this is a hard question, but when I think about the bad effects of instant nooses, sodium is one of them.” (Interviewee no. 50) 

### 4.6. Danish Consumer: Environment-Oriented Customer

#### 4.6.1. Behaviors

##### Theme: Less Meat Consumption

In Denmark, an environment-oriented customer segment tends to consume less meat and more plants to help the environment. They are concerned about soil, air, water pollution, and greenhouse gas emissions from industrial livestock production. Therefore, they decided to eat fewer meats to reduce their environmental impact. Ten of the respondents replied as follows:

“Nowadays, I tend to eat much more vegetables and vegan diets because I am more concerned about the environmental impacts of food production. So, when I buy instant noodles, I will consider the ingredients first.” (Interviewee no. 51) 

“I am trying to cut out meat from my diet and I try to be a flexitarian. Hence, I rarely eat meat now. Also, I like instant noodles, but I probably read the ingredients on the labels before I decided to buy them.”  (Interviewee no. 52)

“Since I study environmental engineering, I tend to be more environmentally conscious. Right now, I’m trying as much as I can to eat less meat. However, I often eat instant noodles while I am at home. Mainly, I buy plant-based instant noodles because I am fond of vegetables and want to eat less meat as well.” (Interviewee no. 53) 

“I am sort of concerned about environmental effects at a certain point. So, I select the eco-friendly diets from the green food shops.” (Interviewee no. 54)

“I have been a vegetarian for 4 years because I realize that eating meat is bad for the environment. I often eat instant noodles when I am on the train. I only put the vegan sausages on the top of instant noodle soups.” (Interviewee no. 55)

“I want sustainable foods that do not harm the environment. So, I started eating less meat due to greenhouse gas emissions.” (Interviewee no. 56)

“To minimize adverse consequences to the environment, I try to consume environmentally friendly foods.” (Interviewee no. 57)

“I stopped eating meat, but I still eat fish. I probably consider myself a pescatarian. On busy days, I sometimes eat instant noodles.” (Interviewee no. 58)

“I’m trying to eat less meat at first because it’s healthier for me. However, I just realized what I am doing is good for the environment.” (Interviewee no. 59)

“I tend to choose food products that have a lower negative impact on the environment. I am not sure if instant noodles are good for your health and the environment or not.” (Interviewee no. 60)

#### 4.6.2. Desired Outcomes

##### Theme: Eco-Friendly Diets

The outcomes of the study demonstrated that the participants of this segment desired processed foods, helping reduce food’s carbon footprint. Additionally, they required eco-friendly packaging of instant noodles to minimize environmental issues. Ten of the respondents replied as follows:

“I like the idea of eating the diets and knowing that I am also helping to save the environment. So, I wish to have an eco-friendly diet.”  (Interviewee no. 51)

“To save the planet, I prefer to have sustainable foods.”  (Interviewee no. 52)

“I would like to support foods that reduce carbon footprint emissions.”  (Interviewee no. 53)

“I love to buy eco-friendly instant noodles.”  (Interviewee no. 54)

“I consider the foods’ carbon footprint on the label before making a purchasing decision.” (Interviewee no. 55)

“I select fresh local foods from local farms because I want to reduce my carbon footprint.”  (Interviewee no. 56)

“I think it would be better for the environment. If the instant noodles’ packaging is environmentally friendly.” (Interviewee no. 57) 

“It might be good to produce gluten-free instant noodles that benefit both health and the environment.”  (Interviewee no. 58)

“I would rather prefer to have healthy foods that create sustainable environmental impacts.” (Interviewee no. 59) 

“I would like to see instant noodles in biodegradable plastic packaging, which is good for the environment.”  (Interviewee no. 60)

#### 4.6.3. Pain Points

##### Theme: Less Sustainable Food Choices

The study results revealed that the respondents of this segment had common pain points in terms of fewer sustainable food options in the markets. Ten of the respondents replied as follows:

“I think there are a few choices for me to shop for eco-friendly diets.” (Interviewee no. 51) 

“I love to have convenience foods, but they are less harmful to the environment as well. However, I cannot find any sustainable instant noodles at the moment.” (Interviewee no. 52) 

“The pain point is the difficulty of finding instant noodles that are less harmful to health and the environment.” (Interviewee no. 53) 

“I have a hard time finding eco-friendly convenience foods in supermarkets. It is an obstacle for me.” (Interviewee no. 54) 

“I feel like I do not have many choices for shopping for instant noodles for vegetarians.” (Interviewee no. 55) 

“The problem could be the few organic foods in supermarkets. Moreover, I am not sure if I can trust that those foods are environmentally friendly or not.” (Interviewee no. 56) 

“Food packaging is one of the issues that I am concerned about when I shop for food. I hardly find convenience foods in eco-friendly packaging.” (Interviewee no. 57) 

“My pain point is getting good food with reduced environmental impacts.” (Interviewee no. 58) 

“I struggle with finding ready-to-eat foods that are healthy and environmentally friendly in grocery shops.” (Interviewee no. 59) 

“For me, the difficulty is finding diets that benefit both myself and the planet.”  (Interviewee no. 60)

We used the segmentation matrix to determine the most suitable target segments for sustainable gluten-free instant noodles based on the customers’ responses to questions about their behaviors, desired outcomes, and pain points. There are five factors: depth of pain, budget, ease of reach, ease of MVP, and market size. Initially, we assessed these five criteria for each section by evaluating interview transcripts. The scores were then categorized as high, medium, or low.

According to Table 3, the convenience-oriented customers in Thailand had a low pain score since they had fewer flavour choices, and the pains were not very harsh compared to the other segments. In contrast, the budget of this segment had a medium score because most of them were high school and university students who earned low incomes. In addition, the ease of reach was high because they often consumed instant noodles daily due to their less time-consuming nature. However, the ease of MVP was evaluated as a medium score for simplicity of making. Ultimately, the market size had a high score because many Thai consumers ate instant noodles.

For the health-oriented customers in Thailand, the depth of pain was high because they had food quality and safety issues with instant noodles. Conversely, the budget was medium because most participants earned less than 1000 USD per month. Additionally, the ease of access was low because they attempted to eat fewer instant foods due to unhealthy ingredients. Nevertheless, the ease of MVP had a medium score for extra benefits from its gluten-free instant noodle prototype. Finally, the market size was rated a medium score because there were an increasing number of health-conscious consumers but still less than other health-unconscious consumers.

For the environment-oriented customers in Thailand, the depth of pain was high because they had product cost pain points regarding eco-friendly food products. The budget of this segment had a medium score since most of the respondents earned less than 1000 USD per month, and the price of sustainable food products was higher than ordinary food products. The ease of reach was low because this segment attempted to consume organic foods from local food shops due to carbon footprint reduction. The ease of MVP was given a medium score for the sustainability of gluten-free instant noodles. Lastly, the size of the market had a low score as there were a small number of Thai consumers who are eco-conscious when purchasing food products.

On the other hand, in Denmark, the convenience-oriented customer had a low score for depth of pain because this segment had issues with the level of spiciness, which is not too severe pain points compared with other segments. In contrast, the budget of this segment had a medium score since most of the participants were university students who earned an average of 895 USD per month, which was not high when compared with other segments. Additionally, the ease of reach was medium because some Danish consumers had eaten instant noodles, but not very often compared with Thai consumers. Moreover, the ease of MVP was a medium score because the instant noodle producers can produce the noodle textures and flavours that satisfy their preferences. Finally, the size of the market had a medium score since instant noodles were popular among young Danish consumers. However, some still eat other Western convenience foods instead of instant noodles. 

The depth of pain was high for health-oriented customers because they encountered problems involving the high sodium content in instant noodles. Furthermore, the budget was high since Danish people earned approximately 5754 USD per month. In contrast, the ease of reach was low because this segment rarely eats convenience foods like instant noodles due to unhealthy ingredients. However, gluten-free instant noodle producers can reasonably produce healthier gluten-free instant noodles to improve digestive systems. Hence, MVP was given a medium score for the benefits of this gluten-free instant noodle. On the other hand, the size of the market has a low score because the number of gluten intolerances among the Danish population was low.

Furthermore, environment-oriented customers have a high degree of pain because they can hardly find eco-friendly instant noodles at the supermarkets. Most of them are making a conscious effort to reduce meat intake in their diet due to the environmental impacts. However, they said that there were not as many options for vegetarian instant noodles in the stores. The budget for this segment had a high score because Danish people had high salaries. In particular, most of the participants in this segment received doctorate degrees with an average annual salary of 97,795 USD. Nevertheless, the ease of reach was medium since some participants attempted to consume fewer convenience foods due to their environmental impact. In addition, instant noodle producers can reasonably produce sustainable, gluten-free instant noodles that meet the requirements of environment-oriented customers. Thus, MVP was given a medium score for the extra benefits of reducing environmental impact. The market size of environment-oriented customers has a high score since green production in Denmark contributes significantly to the Danish economy.

In summary, the segmentation matrix (Table 3) reveals that convenience-oriented customers are the optimal target segment for the sustainable gluten-free instant noodles prototype in Thailand. In contrast, environment-oriented customers are the optimal target segment for the sustainable gluten-free instant noodle prototype in Denmark.

As highlighted in Table 4, the convenience-oriented customer was the potential segment of gluten-free instant noodle products in Thailand. This segment had convenient eating habits, and the most frequent words that appeared in the quotations were “time-saving food preparation”, “regular purchaser of convenience foods”, and “regular users of online food delivery”. In addition, they had demands for sticky and elastic noodles. Most frequently, words that appeared in the quotations were “desire for sticky, elastic, chewy noodles that are difficult to tear into pieces”, “the needs of the soft and smooth textures of noodles”, and “the soft and chewy noodles are easy and fast to cook”. Furthermore, they had common pain points regarding taste dissatisfaction. The most frequent words that emerged were “there are fewer instant noodle brands that meet their taste satisfaction”, “gluten-free instant noodles have fewer flavor options”, and “the unfavourable taste of gluten-free noodles”. 

In contrast, the environment-oriented customer was the potential segment of gluten-free instant noodle products in Denmark. This segment had fewer meat consumption behaviours. The most frequent words that emerged were “they reduce their meat intake by eating more vegetables”, “they became vegetarians and flexitarians”, and “they want to reduce their environmental impact”. Moreover, they preferred eco-friendly diets. The most frequent words were “the needs of eco-friendly processed foods”, “low carbon emissions from the convenience and processed foods”, and “the requirements of eco-friendly food packaging”. Additionally, they shared similar pain points regarding less sustainable food choices. The most frequent words were “there are a few choices of eco-friendly diets in the supermarkets”, “they cannot access sustainable instant noodle products”, and “they are unable to purchase convenience foods with environmentally friendly packaging”. 

## 5. Discussion

The findings indicated that the “convenience-oriented customer” segment is the best target segment for sustainable gluten-free instant noodles in Thailand because they have a higher score in the segmentation matrix than the other two segments. In Thailand, participants in segment 1 were the most likely to consume instant noodles since they did not spend much time preparing food, and they favoured convenience foods such as instant noodles. Hence, convenience was the main reason for consuming instant noodles. This result agrees with the prior study of Mallinson et al. [68]. Moreover, this segment required sticky and elastic textures since they tend to purchase foods based on their food preferences such as texture, taste, appearance, and smell, which is in line with a prior study by Stranieri et al. [69]. This study suggested that individuals purchased convenience foods according to their preferences. In addition, all the participants of segment 1 had common pain points in terms of taste. The results were consistent with the study of sensory perception by Chen and Engelen [70]. This study recommended that flavour (taste and aroma) influence food acceptance and perception. 

In contrast, the “health-oriented customer” segment mainly focused on their nutritional value, health, and quality. They tend to eat fresh (instead of processed) foods and organic (instead of conventional) fruits and vegetables, as they are concerned about the diets that influence their well-being. The result was consistent with the study of buying behaviour towards organic food products by Singh and Verma [71], which suggested that consumers considered health benefits when they purchased organic foods. In addition, this segment needed nutritional value from gluten-free instant noodles as they were highly motivated to improve their health. The finding was consistent with Huang et al. [72], who recommended that nutrition and fitness have become mainstream demands for food consumption among consumers, particularly health-conscious people. Furthermore, participants in segment 2 had common issues regarding food quality and safety since some of them encountered food safety problems and created risks for their health. In particular, most of them were female and completed higher education, and were concerned about food safety and quality, which is consistent with prior research by Haas et al. [73]. Consumers are also more demanding with regard to product quality [74].

On the other hand, Thailand’s “environment-oriented customer” segment changed its eating behaviours because of peer pressure. The finding was consistent with a study by Sangroya and Nayak [75], which demonstrated that peer pressure forced individuals to change their behaviour towards green products. Furthermore, Lee [76] revealed that peer networks generate, circulate, and reinforce an environmental behaviour “norm.” Additionally, this segment demanded more channel shopping for sustainable foods and more eco-friendly diet alternatives in supermarkets. As they stated previously, most of them require various sustainable food options in grocery stores. Hence, they can easily purchase these foods, and not only at specific green supermarkets or restaurants. Furthermore, the respondents of this segment had repeated pain points in terms of high prices for eco-friendly food products, which agrees with Marian et al. [77]. This article suggested that organic food consumers frequently referred to high prices as an obstacle to their purchases. Thus, this obstacle can affect a customer’s purchase intention toward sustainable foods.

On the contrary, the study’s findings demonstrated that the “environment-oriented customer” segment is the best target customer segment for sustainable gluten-free instant noodles in Denmark, since this segment had a higher score in the segmentation matrix than the other two segments. In segment 3, most of them were young women mainly focused on changing food habits by becoming vegetarians due to environmental mitigation regarding meat consumption. Furthermore, people were more environmentally conscious when they made purchases and changed their behaviours, such as eating less meat, avoiding throwing away food, and shopping at locally owned businesses. This result was consistent with a previous study by Dietz et al. [78], which indicated that households changed their behaviour due to carbon emission reduction. Furthermore, this segment demanded convenience foods with co-friendly characteristics since the pro-environmental behaviour of consumers was imposed by the lifestyle in big cities, and they also needed convenience foods due to time constraints. Moreover, this group encountered difficulty accessing more sustainable options, which is consistent with a study by Schösler [79]. In addition, the participants of this segment desired food providers to offer more alternatives to climate-friendly meals in canteens and restaurants. 

Conversely, the “convenience-oriented customer” segment usually purchased convenience foods, ready meals, and takeaway foods since most of them had poor culinary skills and limited time spent in the kitchen. The result was consistent with Hartmann et al. [80], which indicated that convenience food consumption is inversely proportional to cooking ability. Additionally, this segment required the soft texture of instant noodles due to their food texture preferences. Besides, some people do not need much mouth activity but would rather have something to chew and melt in their mouth efficiently [56]. However, they had common problems in terms of the spicy flavour in instant noodles since most traditional Danish dishes are not spicy. Hence, Danish people were not used to eating spicy foods and struggled when eating them.

Lastly, in Denmark, the “health-oriented customer” segment was concerned with nutrition, fitness, and the environment. Therefore, most of them became flexitarians since they were more interested in health issues. In addition, they required nutritional values from instant noodles due to health issues. As previously stated, convenience foods and ready meals typically have low nutritional value. Hence, they rarely eat these foods instead of more organic and natural foods. Many Danish customers also desire food products that maintain a healthy-conscious lifestyle. Moreover, this segment shared a similar pain point regarding the high sodium content in instant noodles, which can put them at a higher risk for health problems.

## 6. Practical Implications

Based on the research findings, the recommendations below were made to three primary stakeholders in Thailand and Denmark: producers, consumers, and policymakers. First, the results of this study suggested that the “convenience-oriented customer” segment was the target customer of sustainable gluten-free instant noodles in Thailand. Accordingly, this study provided the prototype features of gluten-free instant noodles that resolve the pain points of convenience-oriented customers in Thailand. First, the gluten-free instant noodles producers are expected to produce instant noodles that fulfil consumer taste preferences such as spicy, sour, and umami flavours. Second, consumers perceive gluten-free foods as being of inferior quality (particularly sensory quality) compared with traditional (i.e., non-gluten-free) foods [81]. Thus, gluten-free instant noodles producers should produce gluten-free instant noodles with optimal sensory attributes that could satisfy consumers’ perceived or actual needs [82], such as sticky and elastic textures. Moreover, food textures are drivers of food acceptance and perception.

Moreover, convenience-oriented customers are expected to consider taste, nutritional values, and food safety when making food choices. To maintain good health, convenience-oriented customers need to consider essential nutrients and food ingredients’ quality and safety for their health. Consumers should consider food safety certificates or credence attributes (e.g., food certificates and gluten-free certification).

Furthermore, to enhance the quality and safety of instant noodles, policymakers are expected to enforce minimum requirements for nutrients such as protein and control the amount of sodium and MSG so they are not harmful to the consumers’ health. In addition, policymakers should encourage consumers to consume a sufficient amount of nutritious food essential for their body and consume a moderate proportion of certain foods to avoid health conditions such as diabetes, obesity, and low cognitive functioning. Also, Thai policymakers should increase their understanding of gluten-free diets’ health benefits for consumers with celiac and non-celiac gluten sensitivity, since gluten-free diets, including gluten-free instant noodles, can help ease digestive symptoms. Furthermore, policymakers are expected to raise awareness of the environmental impact of food consumption and motivate them to change their eating behaviours in ways that are less harmful to the environment.

In contrast, the findings of our study recommend that the “environment-oriented customer” segment should be the target customer of sustainable gluten-free instant noodles in Denmark. Accordingly, this study provided the prototype features of gluten-free instant noodles that resolve the pain points of environment-oriented customers in Denmark. First, gluten-free instant noodle manufacturers are anticipated to produce more environmentally friendly instant noodles that reduce carbon footprint emissions in production processes such as noodles and packaging. In addition, gluten-free instant noodle producers need to find market solutions to achieve both time-saving and eco-friendly consumer needs. Therefore, the instant noodle manufacturers provide more varieties of instant noodles and other ready-meal options for this customer segment.

Moreover, environment-oriented customers are expected to consider other aspects of food choices such as taste, quality, and health benefits. In addition, consumers should investigate eco-friendly food certifications (e.g., organic certification, green food certification, and non-pollution certification) to ensure that a certain food brand is trustworthy. Moreover, green business certification provides the opportunity to allow consumers to perceive that a particular food company is socially and environmentally conscious with a long-term sustainability plan.

Similarly, policymakers should promote the food industry’s use of cassava in achieving sustainable food security and environmental challenges. Policymakers are expected to encourage food producers and consumers to be aware of health and ecological sustainability in foods, such as cassava roots for sustainable alternative food choices and for gluten-free food options for gluten-intolerant people. Furthermore, policymakers should regulate food producers with regard to carbon footprints and greenhouse gas emissions that are less harmful to the planet. Additionally, policymakers should ensure thatpeople with gluten sensitivity are aware of their diets and health status to avoid severe symptoms.

## 7. Strengths and Limitations of This Study

This is the first published qualitative research utilizing the lean entrepreneur notion proposed by Cooper and Vlaskovits [16] to construct a sustainable gluten-free instant noodle prototype. First, the study highlighted the consumers’ behaviours, desired outcomes, and pain points. Subsequently, the researchers delivered the optimal solutions to alleviate the consumers’ pain points and recommended the solution features for sustainable gluten-free products in Thailand and Denmark.

However, this study has some limitations, but they also present opportunities for future research. First, although the research participants were from different cities across the northeastern Region of Thailand and Odense, Denmark, sampling and data collection were exclusively conducted in the specific region of Thailand and specific cities in Denmark because there were limitations on time and commuting when conducting the study in Denmark. Hence, although the results of this study may have empirical relevance, they should be interpreted with caution. Second, the information provided by this study is insufficient to help the generalized market of other gluten-free food products because we only focused on gluten-free instant noodles, which are specialized in comparison to general foods.

## 8. Conclusions

Instant noodle products are a popular food among consumers in many parts of the world, including Thailand and Denmark. However, instant noodles have been associated with health and environmental impacts due to their unnatural ingredients and production processes. As a result, this study emphasized the use of cassava root as a sustainable alternative food ingredient in gluten-free instant noodles, which aids in achieving long-term food security and addressing environmental challenges. Moreover, gluten-free instant noodles have advantages for gluten sensitivity and non-gluten sensitivity consumers by improving the digestive system. Thus, to develop sustainable gluten-free instant noodle products, this research utilized the lean entrepreneur notion to explore the consumers’ behaviours, desired outcomes, and pain points in consuming instant noodles. Ultimately, the producers can produce instant noodles with product features suitable for their optimal customer segment.

The findings revealed that the “convenience-oriented customer” segment was the target customer segment of gluten-free instant noodles made from cassava in Thailand. First, we found that the convenience-oriented customers often consume instant noodles since they do not spend much time preparing food and favour convenience foods such as instant noodles. Hence, convenience was the main reason for consuming instant noodles. Moreover, this segment required sticky and elastic textures since they tend to focus on food textures when selecting foods. In addition, all the participants in segment 1 had common pain points in terms of taste. Thus, gluten-free instant noodle producers are expected to produce instant noodles that fulfill consumer taste preferences such as spicy, sour, and umami flavours. In addition, companies that make gluten-free instant noodles should make foods with textures that customers want, such as sticky and stretchy ones.

On the other hand, the results demonstrated that the “environment-oriented customer” segment was the target customer of gluten-free instant noodles made from cassavas in Denmark. We found that the environment-oriented customers were primarily young women who focused on changing food habits by becoming vegetarians due to environmental mitigation regarding meat consumption. Furthermore, people were more environmentally conscious when they made purchases and changed their behaviours, such as eating less meat, avoiding throwing away food, and shopping at locally owned businesses. In addition, this segment demanded convenience foods with eco-friendly characteristics since the pro-environmental behaviour of consumers was imposed by the lifestyle in big cities, and they also needed convenience foods due to time constraints. Moreover, this group encountered difficulty accessing more sustainable options and wanted food providers to offer more alternatives to climate-friendly meals in canteens and restaurants. Therefore, gluten-free instant noodle manufacturers are anticipated to produce more ecologically friendly instant noodles that diminish carbon footprint emissions in production processes such as noodles and packaging. In addition, gluten-free instant noodle producers need to find market solutions to achieve both time-saving and eco-friendly consumer needs. Therefore, instant noodle manufacturers should provide more varieties of instant noodles and other ready-meal options for this customer segment.

In conclusion, this study gives insight into Thai and Danish customers’ behaviors, desired outcomes, and pain points toward instant noodle consumption. It also identifies the most promising customer segment for gluten-free instant noodles in Thailand and Denmark. These include highlighting the novel use of sustainable food ingredients, like cassava, in the food industry. As a result, research and development projects to improve gluten-free instant noodle products made from cassava root are critical for the successful consumer adoption of these products. 

## Figures and Tables

**Figure 1 foods-11-02437-f001:**
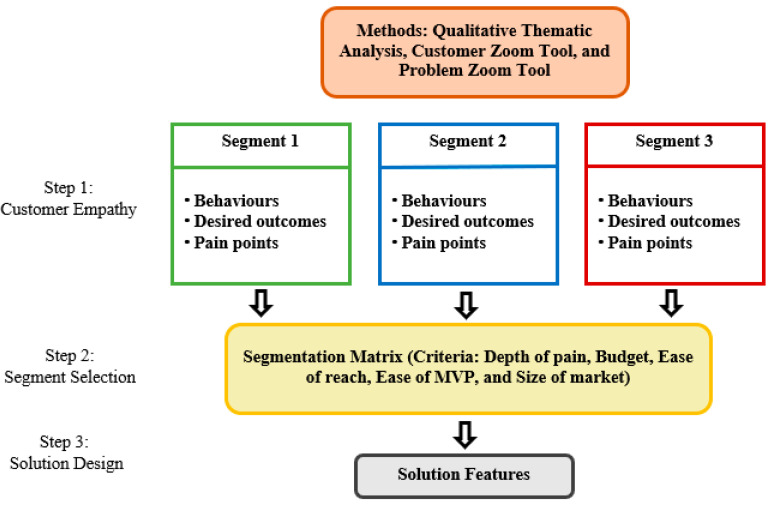
Theoretical Framework. Source. Figure created by authors (2022).

**Figure 2 foods-11-02437-f002:**
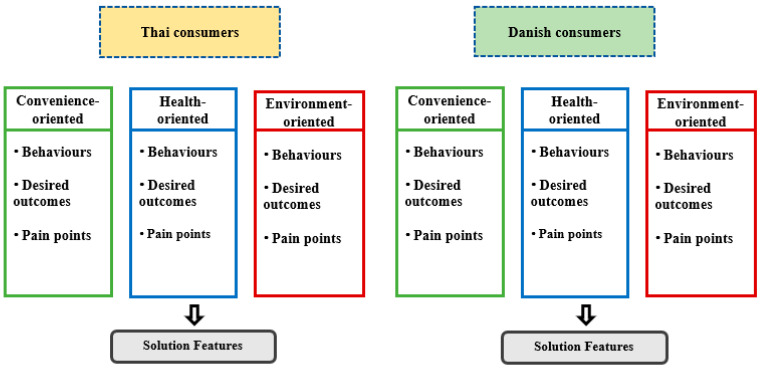
Research Framework. Source. Figure created by authors (2022).

**Table 1 foods-11-02437-t001:** Segmentation matrix used to classify consumers into the three groups (convenience-oriented, health-oriented, and environment-oriented).

Customer	Depth of Pain	Budget	Ease of Reach	Ease of MVP	Size of Market	Total Scores	Rank
Segment 1	H, M, L	H, M, L	H, M, L	H, M, L	H, M, L		
Segment 2	H, M, L	H, M, L	H, M, L	H, M, L	H, M, L		
Segment 3	H, M, L	H, M, L	H, M, L	H, M, L	H, M, L		

Source. Data created by authors (2022). Note. H = high; M = medium; L = low.

**Table 2 foods-11-02437-t002:** Demographic Profiles of Interview Participants (*n* = 60).

Demographic Variable	Category	Number of Participants					
		Three Customer Segments					
		Convenience-Oriented Customer		Health-Oriented Customer		Environment-Oriented Customer	
		Thai (*n* = 10)	Danish (*n* = 10)	Thai (*n* = 10)	Danish (*n* = 10)	Thai (*n* = 10)	Danish (*n* = 10)
Gender	Male	4	5	3	2	3	4
	Female	6	5	7	8	7	6
Age (years)	20–30	5	8	0	2	3	5
	31–40	2	2	1	4	4	4
	41–50	3	0	3	3	1	1
	51–60	0	0	6	1	2	0
Education level	Secondary education	1	0	0	0	0	0
	Vocational certificate/diploma	2	0	0	0	2	0
	Bachelor	5	4	5	3	4	2
	Master	2	4	2	3	2	3
	PhD	0	2	3	4	2	5
Occupation	Student	6	8	2	3	2	5
	Public sector employee	2	1	4	3	1	2
	Private sector employee	2	0	2	4	5	2
	Business owner	0	0	2	0	2	0
	Others	0	1	0	0	0	1
Income (USD)	Less than 1000	9	7	6	0	1	0
	1001–2000	1	2	1	0	3	0
	2001–3000	0	1	2	1	4	1
	3001–4000	0	0	1	2	1	1
	4001–5000	0	0	0	5	1	2
	More than 5000	0	0	0	2	0	6
Purchase frequency (per week)	Less than once	1	9	4	10	5	10
	1–2 times	2	1	3	0	2	0
	3–4 times	5	0	2	0	2	0
	More than 4 times	2	0	1	0	1	0

Source. Data adapted from authors (2022).

**Table 3 foods-11-02437-t003:** Segmentation matrix of Thai and Danish consumers.

Nationality	Customer	Depth of Pain	Budget	Ease of Reach	Ease of MVP	Size of Market	Total Scores	Rank
Thai	Convenience-oriented customer	L	M	H	M	H	11	1 ***
	Health-oriented customer	H	M	L	M	M	10	2
	Environment-oriented customer	H	M	L	M	L	9	3
Danish	Convenience-oriented customer	L	M	M	M	M	9	3
	Health-oriented customer	H	H	L	M	L	10	2
	Environment-oriented customer	H	H	M	M	H	13	1 ***

Source. Data adapted from authors (2022). Note. H = high; M = medium; L = low. Note. *** Implying the highest possible scores.

**Table 4 foods-11-02437-t004:** Summary of the potential segment from each country.

Nationality	Segment	Behaviours	Desired Outcomes	Pain Points
Thai	Convenience-oriented customer	Convenient eating habits	Sticky and elastic noodles	Taste dissatisfaction
		Time-saving food preparation	The desire for sticky, elastic, chewy noodles that are difficult to tear into pieces	There are fewer instant noodle brands that meet their taste satisfaction
		The regular purchaser of convenience foods	The need for the soft and smooth textures of noodles	Gluten-free instant noodles have fewer flavor options.
		Regular users of online food delivery	The soft and chewy noodles are easy and fast to cook.	The unfavourable taste of gluten-free noodles
Danish	Environment-oriented customer	Less meat consumption	Eco-friendly diets	Less sustainable food choices
		They reduce their meat intake by eating more vegetables	The need for eco-friendly processed foods	There are a few choices of eco-friendly diets in the supermarkets
		They became vegetarians and flexitarians.	Low carbon emissions from the convenience and processed foods.	They cannot access sustainable instant noodle products
		They want to reduce their environmental impact	The requirements of eco-friendly food packaging	They are unable to purchase convenience foods with environmentally friendly packaging

Source. Data adapted from authors (2022).

## Data Availability

Not available.

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
