# Peer review of "Conceptualizing a Gluten-Free Instant Noodle Prototype Using Environmental Sustainability Aspects: A Cross-National Qualitative Study on Thai and Danish Consumers"

_foods, 2022, doi:10.3390/foods11162437_

Round 1

Reviewer 1 Report

This paper conceptualizes a gluten-free instant noodle prototype by using semi-structured interviews in Thailand and Denmark. Overall, I think this is an interesting work that adds interesting points to consumer behaviour literature. The paper, despite being well written, should be better organized in order to enhance readability and to stimulate readers’ interest. My major concern is with the rationale of this paper which does not come out very clearly. Please see my comments below. I hope it contributes to guiding you through your revision.

Introduction and Literature review

Both these sections should be better organized and written.

I can follow the narration flow, but it has to be improved. There are many sentences not corroborated by data (see for example lines 52-53; 60 -67) or without a logical connection (see for example lines 58 to 60).

Another issue sits in the rationale behind the choice of gluten-free instant noodle as the main object of the study (see lines 68-73).

Furthermore, the explanation of NOT choosing well-established techniques is very weak and without references (see lines 87-92).

Also lines 108-110 looks like an advertisement and is not suitable for a scientific journal.

At line 117 authors provide an explanation of what we have already read, please remove the description of the introduction.

The introduction section should be better focussed on gluten free products, including instant noodles by pointing out pros and cons from both stakeholders’ and consumers’ perspective. I recall this is addressed in the literature review but, in my opinion, this should be completely revised.

Research Methodology

This whole section is well written and very easy to read. My only concern sits in lines 345-372 where we have a repetition of all the questions and that I suggest to remove since there are appendices A.1 and A.2.

Results

Tables 3 to 8 are almost useless, please put them in appendices or remove them. Furthermore, I would have preferred a more narrative version of paragraphs (and subparagraph) 4.1 to 4.6.3., as it starts from line 943 which is very easy to read and follow.

Author Response

Thank you very much for your review and comments. Here are our responses.

Reviewer 1

This paper conceptualizes a gluten-free instant noodle prototype by using semi-structured interviews in Thailand and Denmark. Overall, I think this is an interesting work that adds interesting points to consumer behaviour literature. The paper, despite being well written, should be better organized in order to enhance readability and to stimulate readers’ interest. My major concern is with the rationale of this paper which does not come out very clearly. Please see my comments below. I hope it contributes to guiding you through your revision.

Introduction and Literature review

Both these sections should be better organized and written.

I can follow the narration flow, but it has to be improved. There are many sentences not corroborated by data (see for example lines 52-53; 60 -67) or without a logical connection (see for example lines 58 to 60).

Another issue sits in the rationale behind the choice of gluten-free instant noodle as the main object of the study (see lines 68-73).

Response:

We decided to erase the irrelevant statements. Line 52-56 “In Thailand, instant noodles have become popular among Thai consumers. Hence, noodles producers attempt to produce local instant noodles.”

For line 58-60, line 60-67, and line 68-73, please see the pink highlights on page 4 paragraph 2, 3, and 4.

Furthermore, the explanation of NOT choosing well-established techniques is very weak and without references (see lines 87-92).

Response: Please see the last paragraph on page 2 and the first paragraph on page 3.

Also lines 108-110 looks like an advertisement and is not suitable for a scientific journal.

Response: This statement was corrected. See the highlight on page 3 paragraph 1.

At line 117 authors provide an explanation of what we have already read, please remove the description of the introduction.

Response: We removed line 117-124 (based on the comments of reviewer 2).

The introduction section should be better focussed on gluten free products, including instant noodles by pointing out pros and cons from both stakeholders’ and consumers’ perspective. I recall this is addressed in the literature review but, in my opinion, this should be completely revised.

Research Methodology

This whole section is well written and very easy to read. My only concern sits in lines 345-372 where we have a repetition of all the questions and that I suggest to remove since there are appendices A.1 and A.2.

Response: The questions were removed and we suggest seeing the appendix section instead.

Results

Tables 3 to 8 are almost useless, please put them in appendices or remove them. Furthermore, I would have preferred a more narrative version of paragraphs (and subparagraph) 4.1 to 4.6.3., as it starts from line 943 which is very easy to read and follow.

Response: Table 3-8 were moved to Appendix 2. Line 432-444 were removed.

Reviewer 2 Report

This study aims to identify the target consumer segments and suggest new formulations for gluten-free instant noodles. The introduction, objectives, methods and discussion are well-written. The main drawback of this study is the results section, the study is quite large and there a lot of results but the results can be quite tedious to the reader as it is currently constructed. The authors may want to consider trying to streamline the results section or only highlighting major trends. Other comments are listed below:

Line 85-89: The authors state that focus groups are used in the study, but then in the proceeding statement sat these do not address the current research objectives. I think these statements should be switched or clarified for the reader.

Line 117- 124: I do not think this paragraph is necessary.

Line 176- “Nutritional information on yoghurt’s acceptability…”, this statement is confusing.

Line 432-444: Is very repetitive and it may not be necessary to include this information. Maybe, include it in the appendix? Along with the tables 3-8.

Line 1121- “In addition”, is used frequently throughout this section. The authors may want to use other words to increase the readability of the section.

Author Response

Thank you very much for your review and comments. Here are our responses.

Reviewer 2

This study aims to identify the target consumer segments and suggest new formulations for gluten-free instant noodles. The introduction, objectives, methods and discussion are well-written. The main drawback of this study is the results section, the study is quite large and there a lot of results but the results can be quite tedious to the reader as it is currently constructed. The authors may want to consider trying to streamline the results section or only highlighting major trends. Other comments are listed below:

Line 85-89: The authors state that focus groups are used in the study, but then in the proceeding statement sat these do not address the current research objectives. I think these statements should be switched or clarified for the reader.

Response: Corrected. See the pink highlights on page 2.

Line 117- 124: I do not think this paragraph is necessary.

Response: We removed line 117-124. See page 3.

Line 176- “Nutritional information on yoghurt’s acceptability…”, this statement is confusing.

Response: Corrected.  See pink highlights on page 4.

Line 432-444: Is very repetitive and it may not be necessary to include this information. Maybe, include it in the appendix? Along with the tables 3-8.

Response: Table 3-8 were moved to Appendix 2. Line 432-444 were removed. See page 10.

Line 1121- “In addition”, is used frequently throughout this section. The authors may want to use other words to increase the readability of the section.

Response: Many parts were corrected. See pink highlights on page 24 and 25.

Round 2

Reviewer 1 Report

I would like to thank the authors for following my suggestions.